# Understanding the Implementation of Antimicrobial Policies: Lessons from the Hong Kong Strategy and Action Plan

**DOI:** 10.3390/antibiotics11050636

**Published:** 2022-05-10

**Authors:** Mingqi Song, Ziru Deng, Olivia Chan, Karen Ann Grépin

**Affiliations:** School of Public Health, University of Hong Kong, Hong Kong, China; msong97@hku.hk (M.S.); dengziru@connect.hku.hk (Z.D.); oliviachan@hku.hk (O.C.)

**Keywords:** antimicrobial resistance, policy implementation, Hong Kong, national action plan, One Health

## Abstract

In 2017, the *Hong Kong Strategy and Action Plan on Antimicrobial Resistance 2017–2022* (HKSAP) was announced with the aim of tackling the growing threat of antimicrobial resistance (AMR) in Hong Kong. However, little is known about how the planned activities have been implemented. In this study, we examine the status of implementation of the HKSAP using the Smith Policy Implementation Process Model. Semi-structured interviews with 17 informants found that important achievements have been made, including launching educational and training activities targeting the public, farmers, and healthcare professionals; upgrading the AMR surveillance system; and strengthening AMR stewardship and infection control. Nevertheless, participants also identified barriers to greater implementation, such as tensions across sectors, ongoing inappropriate drug use and prescription habits, insufficient human and technical resources, as well as a weak accountability framework. Environmental factors such as the COVID-19 pandemic also affected the implementation of HKSAP. Our study indicated that expanding engagement with the public and professionals, creating a collaborative environment for policy implementation, and building a well-functioning monitoring and evaluation system should be areas to focus on in future AMR policies.

## 1. Introduction

Recent decades have witnessed the rise of antimicrobial resistance (AMR) as a major health crisis. Worldwide, AMR itself directly caused 1.27 million deaths and was associated with 4.95 million deaths in 2019, becoming a leading cause of mortality. Regionally, according to a recent analysis, the AMR burden was particularly high in sub-Saharan Africa and South Asia [1]. It was estimated that by 2030, up to 24 million people will live in extreme poverty due to AMR [2]. According to the World Bank, AMR will expand the inequity between countries and impose the largest shortfalls in economic growth in low-income countries [3]. Thus, combating AMR has been a global health priority for over two decades.

Following the lead of the World Health Organization (WHO), many countries and regions introduced action plans to mitigate AMR according to the WHO’s Global Action Plan (GAP) [4]. As of 2019, 117 countries and regions had approved AMR national action plans (NAP) [5]. While most of these are aligned with the GAP framework, in practice, critical gaps exist concerning the implementation of the NAPs. According to the Interagency Coordination Group (IACG) discussion paper on AMR NAPs, implementation and sustained policy attention, instead of NAP design, are the main challenges in the majority of countries. Awareness, political will, financing, coordination, monitoring, and data, along with technical capacity, are additional important challenges to the implementation of NAPs [6]. Systematic reviews have further identified multi-level collaborations, laboratory practices, and data capture as other important determinants of successful NAP implementation [7,8].

Several country-level case studies have also investigated the factors hindering AMR policy implementation. In countries at different levels of development (e.g., Singapore, Pakistan, Bangladesh), the lack of AMR awareness among the public, together with limited community engagement, were identified as primary implementation challenges [9,10,11]. Other widely reported issues include the insufficient participation and low status of non-human sectors (i.e., animal and environmental health) [9,11,12], which have subsequently resulted in inadequate AMR surveillance and poor resource mobilization. Lax policy implementation in the private sector has also been noted as a common problem in many countries as well [7,9,13,14].

However, policy implementation is also likely context-specific, and there is “unlikely to be a ‘silver bullet’ intervention that solves the AMR problem” [15]. In Singapore, the need for dedicated funding has been regarded as critical for successful AMR policy implementation [9]. Furthermore, a lack of predictable funding was revealed to be an important challenge in Pakistan, Tanzania, and Bangladesh [10,11,16]. In Thailand, opposition from those who disagreed with antibiotic reclassification impeded the implementation of antimicrobial regulations [12]. In Iran, external factors, such as knowledge transfer, advocacy for policy agenda-setting, and technical support, have been shown to shape domestic policy implementation [17]. Fragmented policy implementation at the local level has been identified in other contexts. For example, in China, antimicrobial use (AMU) surveillance was found to be less adequate in rural areas and primary healthcare settings than in urban areas and other healthcare institutions [18]. 

Hong Kong’s policy efforts to tackle AMR can be traced back to the *Antibiotics Ordinance*, a law passed in 1948 to regulate the sale and possession of antibiotics [19]. The AMR landscape in Hong Kong has undergone several changes from 2005 to 2017. In 2005, a working group on antimicrobial stewardship (AMS) was established under the Centre for Health Protection (CHP) [20], which led to the development of AMS programs at public hospitals [21]. In 2016, the High-Level Steering Committee on AMR (HLSC) was assembled. Later, the Expert Committee on Antimicrobial Resistance (EC) was formed to provide authoritative opinions to the HLSC. In 2017, the government launched the *Hong Kong Strategy and Action Plan on Antimicrobial Resistance 2017–2022* (HKSAP), its version of an AMR NAP. In May 2021, the HKSAP Mid-Term Review Report was published [22], detailing the progress made along with the major challenges faced since 2017.

Despite increased policy attention, AMR rates remain high in Hong Kong, exceeding those of many developed economies [23]. The increasing resistance trends have primarily been observed in public hospitals [24]. Moreover, AMR transcends the human health sector, and it also affects the animal health and food sectors. As of 2019, audit tests on 168 pig urine samples collected from local farms found antimicrobial residues in 33 samples [25]. AMR has also become a food safety concern. In June 2020, 46 of 304 (15.1%) ready-to-eat-food samples tested positive for at least one of the three major resistant organisms [26]. It is therefore unclear if the policies that have been adopted thus far are effective or sufficient in controlling AMR in Hong Kong.

The implementation of AMR policies in Hong Kong remains underinvestigated. Previous studies have investigated the knowledge, attitude, and practice of the Hong Kong people towards antibiotics in conjunction with antibiotic-dispensing patterns, the characteristics of consumers, and the effectiveness of awareness-raising programs [27,28,29,30]. Ogyu et al. compared the content laid out in the HKSAP with the GAP [31]. However, there have been no formal studies of AMR policy implementation in Hong Kong and the challenges and barriers to greater policy implementation. Against this background, the objectives of this paper are twofold: first, to explore the status of the implementation of the HKSAP, especially the motivations, contextual drivers, and interactions among policy actors; second, to examine the challenges associated with implementing AMR policies in the Hong Kong context. Further, as a developed international trading metropolis, understanding Hong Kong’s efforts and challenges in combating AMR would shine new light on the existing understanding of the HKSAP while also offering insights for policymakers of other comparable settings.

## 2. Materials and Methods

### 2.1. Study Design

Numerous frameworks have been developed to describe, analyze, and guide policy implementation [32,33,34]. In the context of AMR, one notable framework is the interdisciplinary AMR-Intervene Framework designed by Leger et al., which contains six components of interventions [35]. Another is the Normalization Processing Theory (NPT), a framework to assess and enhance complex interventions [36]; this specific framework was used by Currie et al. to analyze mechanisms affecting the implementation of a national AMS program in Scotland [37]. 

The framework that is of particular interest for this paper is the Smith Policy Implementation Process Model (Figure 1). Introduced in 1973, the Smith Model identifies four essential components of the policy implementation process: idealized policy (the idealized patterns of interaction that the policymakers wish to induce), target groups (people who are affected by and required to adapt to new patterns introduced by the policy), implementing organizations (agencies responsible for policy implementation), and environmental factors (influencing policy implementation) [38,39]. The interaction within and between these components produces tensions and fosters or hinders policy implementation through feedback, leading to further changes in policies [40]. 

We selected this model as the analytical framework for this study as it underscores the importance of viewing policy implementation in its context and as an ongoing process. It provides a lens with which to examine the context, implementors, as well as dynamic interactions among stakeholders. It sheds light on how the interactions across different sectors would affect the effectiveness of policies and further impact policy change. 

To address our specific research questions, we made the following adjustments to the original Smith Model: (1) instead of describing the contents of the idealized policy, we presented informants’ views on the HKSAP; (2) “Transactions” and “Institutions” were merged into “Feedback”, as there is overlap between the three components; (3) “Tensions” was rephrased as “Interactions and tensions” when discussing both the positive and negative sides of the implementation process; (4) for “Implementing organizations”, we focused on “the implementing programs and capacity”; (5) for “Environmental factors”, we only discussed factors that influenced the implementation of the HKSAP and did not explore how these factors were affected by HKSAP.

### 2.2. Data Collection

Potential interviewees were selected by first identifying implementing organizations and the individuals involved in the execution of the HKSAP. We constructed a list of additional respondents who likely had some roles in the implementation process by using the telephone directory of the Hong Kong government and reviewing the organizational structure of related government departments. We also took note of individuals named on academic papers, policy documents, and websites of professional associations. The tailored invitation emails were sent to all potential interviewees. In total, 17 out of 116 key informants that were targeted agreed to be interviewed. Among them, two participants had not been on the initial list of interviewees but were later identified by another interviewee.

The semi-structured interviews were conducted from April to October 2020. The interview guidelines were adapted from an interview guide developed to study AMR policy implementation in a multi-country setting. While there was a long list of potential questions (Table 1), we modified the interview guide based on each informant’s background, sector, and level of engagement with the HKSAP. All interviews were conducted online in English via Zoom due to the COVID-19 pandemic, except for one, which was conducted in person.

Table 2 summarizes the characteristics of the study participants. Two participants were members of the HLSC, and four were members of various committees set up by government departments responsible for the application of the HKSAP, including an informant from EC.

### 2.3. Data Management and Analysis

All the interviews were recorded, transcribed, and cleaned verbatim using Otter.ai. Two researchers designed a codebook based on the modified Smith Model and the research questions. They then combined and compared all selected quotes and generated a collated list of themes via QSR NVivo 12. 

## 3. Results

### 3.1. Idealized Policy

The respondents generally agreed that the HKSAP is comprehensive in that it targets the primary drivers of AMR in Hong Kong. The plan also received praise from many informants due to its usefulness in framing the overall direction of AMR policies. 

However, the feasibility of achieving the goals set out in the HKSAP was questioned by many informants. Two informants felt that the content of the HKSAP was too generic, thereby diminishing the value of its accountability framework. As one informant commented, “*the HKSAP is based on what has been done regularly … The goals will be applicable forever. But in any case, it does not matter because we will just continue in the same direction*” (KI1, DH official). Another respondent thought there was an excessive focus on “low-hanging fruits” or problems that are easier to address (KI8, human health).

A few informants further reflected on possible reasons for the reported issues in the HKSAP. For example, the nonexistence of “*an integrated surveillance system before the HKSAP was launched*” (KI1, DH official) could be a potential reason for some of the design flaws, meaning that there had been insufficient data available to policymakers during the policy-formulation phase. Moreover, many informants believed that patients and other relevant stakeholders had not been sufficiently involved during policy formulation. As one informant noted, “*the Action Plan genuinely missed the anthropological input. It is important to understand the constraints and concerning issues that real people from the ground level are facing day to day*” (KI3, animal health). 

Another contributing factor was the imbalance of power across sectors in driving the direction of the HKSAP. One respondent from livestock farming said, “*the food animal and farming industries are still very weak. We do not have many negotiating powers*” (KI9, animal health). This juxtaposes comments from an interviewee from the government who opined, “*government try to set up sustainable AMR offices and policies*” (KI2, DH official). The contrasting views describe a discourse in implementation power alignment, which might affect resource allocation and impair the feasibility of planned activities as well as goals assigned to the less-powerful sectors. Some informants were also frustrated that too little attention had been given to the environmental health sector. “*In the action plan, you can see that they focus on the human side, animal side and food side, but not environmental*” (KI8, human health). 

### 3.2. Implementing Organizations

The involvement of agencies and actors from different sectors and bureaucracy levels is central to the successful implementation of the HKSAP. Our interview data demonstrated that AMR surveillance, educational programs for both the public and professionals, as well as AMS in hospital settings, were being effectively implemented. However, suboptimal intersectoral and multisectoral coordination in data collection, limited enforcement of AMU regulations, and insufficient resources were observed.

#### 3.2.1. Human Health Sector

In this sector, activities had mainly focused on data and surveillance, education, and training, along with hospital practices, to control inappropriate AMU. The Department of Health (DH) and the Hospital Authority (HA) have made important efforts to upgrade the surveillance system. The level of data reporting and categories of key indicators in the HKSAP were also more detailed compared to the "pre-HKSAP" era. In parallel, the prescription review, diagnostics surveillance, and lab testing were continued under the hospital-based AMS program. In community pharmacies, irregular test purchases and advocacy activities for appropriate AMU were implemented as well (KI5, human health). 

Education efforts had been deemed to be particularly well implemented. Since 2017, the CHP has developed tailored educational material for different audiences, and this is indicated as “Key area 4: communication, training, and education” activities in the HKSAP. AMR was also added to the Diploma of Secondary Education (DSE) and the Continuing Medical Education Program (CME) as necessary material (KI2, DH official). At the same time, the HA launched an e-learning project on antibiotic prescription and “*made it a mandatory requirement for interns*” because the HA management team believed that “*it will be better to educate young doctors*” (KI11, human health). Furthermore, together with other stakeholders, the CHP formulated guidelines on AMU for commonly encountered infections in primary care settings. These guidelines, as one informant remarked, “*have been promulgated among primary care physicians and received very good feedback*” (KI2, DH official). 

Our interviews identified three major implementation challenges in this sector. First, data collection efforts were largely fragmented. It was considered time-consuming to sort and reorganize data flows because the data extracted from hospitals were for clinical purposes rather than surveillance (KI14, human health). The data standards adopted by hospitals were also highly variable. Second, it was difficult to enforce AMU policies due to the inability to directly observe the interactions between patients and providers: “*Sometimes we judge it as inappropriate or misuse [of antibiotics]. The doctor may find that they have some reasons to prescribe the drugs. So, it is quite difficult to judge*” (KI2, DH official). Third, owing to the dual-track nature of Hong Kong’s healthcare system, collecting antibiotic usage data from private doctors was a major challenge, “*especially for those solo practice offices*” (KI15, human health). On top of that, the CHP’s engagement with the private and non-government sectors was deemed insufficient. This not only led to a low level of awareness of the HKSAP among these stakeholders but further limited the execution of key policies.

#### 3.2.2. Food and Animal Health Sectors

There have been surveillance efforts for both AMU and AMR in the food-producing animal sector (KI3, animal health; KI12, animal health). The Agriculture, Fisheries and Conservation Department (AFCD) was implementing alternatives to the direct purchase of antimicrobials so that “*farmers could no longer directly purchase antimicrobials and instead must get antimicrobials through veterinarians*” (KI3, animal health). To better meet the demand of farmers for veterinary services, the Centre for Food Safety (CFS) contracted the City University of Hong Kong (CityU) to establish a new dispensing service. An informant described the workflow: “*When our farm has a disease problem, we request the veterinarian from the CityU to give us prescriptions of drugs. They only give you a reasonable amount*” (KI9, animal health). The CityU was also responsible for collecting data and monitoring AMU conditions among farmers (KI3, animal health).

The key challenges reported in this sector include the lack of technical support for appropriate AMU, limited veterinary services, and gaps in the data collection process. Although the AFCD contracted with the CityU to develop and provide veterinary services for local farms, informants opined that this support was insufficient to carry out all surveillance activities laid out in the HKSAP (KI9, animal health). Furthermore, as far as data collection was concerned, reporting AMU data was not mandatory for farmers. The accuracy and completeness of reported AMU data was questioned by farmers: “*I do not know whether these diagrams and tables are meaningful because there are definitely some people who just give out fake figures*” (KI9, animal health).

Informants also underscored how the sluggish process of farm modernization had blocked the implementation of AMR policies in these sectors. Modern farms are more conducive to effective infection control, but current regulations place limits on the height of the local farmhouses, which is “*not good for health because of the poor ventilation*” (KI10, animal health). Another informant said that “*if the government asked us to minimize AMU, they must let us modernize the farm*” (KI9, animal health). Thus far, little support has been given to farmers to help them upgrade their farms. Additionally, the applications of agricultural structure construction, which would allow for such upgrades, can only be referred to the Lands Department for direct handling. This process has been viewed as inefficient from the perspective of farmers. As an informant summarized, “*it is more like a legislation regarding farming structure problem instead of an antimicrobial-related scientific problem*” (KI9, animal health). 

#### 3.2.3. Other Challenges

Informants from all sectors agreed that the resources available were inadequate for implementing the HKSAP. An informant working in the hospital setting emphasized that “*the major hurdle in the implementation of the programs in hospital is human resource. There are not enough pharmacists, trained microbiologists, and infectious disease specialists to do the adults*”. There could even be “*a fight for manpower*” among hospitals (KI6, human health). Other measures of the infection-control procedure could be affected by the shortage of human capital as well. Another informant stressed the need for information technology to monitor AMU instead of deploying greater manpower (KI11, human health). In terms of financing the HKSAP’s implementation, an informant from the animal health sector was concerned about the sustainability of veterinary services due to limited funding. “*The service is free now, but the CityU does have a tight budget. For example, I had a disease outbreak two months ago. And we could only do a post-mortem of two pigs. They have to regulate the number of services they can offer*” (KI9, animal health).

Compared to other sectors, AMR interventions in the environmental health sector were still nascent. An informant felt that “*the government do not really pay attention on the problem of antibiotics in the environment*” (KI8, human health).

### 3.3. Target Groups

The general public, medical practitioners, and farmers emerged as the three major targets of the HKSAP. The challenges associated with these target groups include resistance to AMU behavior change, lack of public trust in implementing agencies, doctors’ autonomy over prescriptions, and existing loopholes in cross-border trade policies.

To better understand the “*different sets of medical doctors or the general public’s ideas on AMR*”, surveys were regularly conducted to collect information about knowledge, attitudes, and practices towards AMR (KI2, DH official). The target groups were also invited to join various educational activities designed for their respective sectors, and these received positive feedback. One animal health participant recalled that farmers were given training on “*what is the proper use of antimicrobials and how bad the situation is*”, after which they began to record the types and volume of antimicrobials they used (KI9, animal health). 

While medical practitioners have been on the receiving end of some educational activities, they have also played an instrumental role in educating the public about appropriate AMU. According to one informant, “*[nurses and doctors’] contributed quite a lot in the publicity events and the preparation of publicity material to increase the awareness of the AMR*” (KI2, DH official). WhatsApp groups were created and served as important communication channels between community pharmacies and governmental agencies, through which government officials reminded pharmacists of the need for responsible use of antibiotics (KI5, human health). 

Despite the progress achieved so far, the implementation of AMU regulations and surveillance as planned in the HKSAP continue to be hampered by the knowledge and attitudes of the target groups. Among the general public, “*health literacy is not high in Hong Kong. There are still a lot of people who cannot tell what kind of drugs they are taking. […] They mainly rely on the advice of their doctors*” (KI4, human health). A related issue was patients’ beliefs: “*The public believes that [using antibiotics] is a very convenient way to remove the symptoms of the diseases. Sometimes, they even go to the community pharmacy to purchase antibiotics without a prescription*” (KI8, human health). The propensity to avoid doctor consultation may be driven by financial factors. “*When they see a doctor, they need to pay more. […] So, although it is illegal to get antibiotics without a prescription over the counter, there are still many people who will try to violate the law*” (KI2, DH official). 

Similar sentiments were also identified among livestock farmers who appeared to be resistant to change. As farmers have been self-reliant and followed certain common industry practices for decades, they found it difficult to put their trust in professional technicians appointed by the government. As one informant put it, “*our industry has grown up without this professional input. It is very difficult for farmers to accept that there is something to provide for them*” (KI3, animal health). Additionally, AMR surveillance and many other planned interventions stated in the HKSAP were relatively new to local farmers. An informant opined that “*most farmers are not educated. They have not been taking any records since day one of pig farming. If you ask them to write down how many milligrams they use, it is a pain*” (KI9, animal health). This could also explain why there was an underreporting issue when it came to AMU data collection, as compliance with data reporting policies was not simple for local farmers. 

Additionally, while physicians in Hong Kong have a high level of professionalism and significant autonomy when prescribing drugs, they are also “*quite famous for prescribing antibiotics for patients for very minor symptoms*” (KI9, animal health). Additionally, such defensive prescription habits are not easy to change: “*they just have their own wheels […] because they think that broad-spectrum antimicrobial stewardship to their patients even without explanation*” (KI11, human health). In some cases, patients have also pressured doctors to adopt a defensive approach to medication, hindering the enforcement of AMU regulations in both public and private healthcare settings. “*When doctors are treating the patients, they are afraid that patients may not appreciate that you are working for the public good…because patients tend to sue you if they think you are not giving them the best [drugs], especially in the private sector*” (KI14, animal health). 

Lastly, the purchase of illegal antibiotics posed another challenge for the implementation of AMU-related interventions. These behaviors are hard to eliminate due to the loopholes in cross-border trade policies and the relentless pursuit of profit. According to an animal health informant, “*Hong Kong farmers are smart in the way they use antibiotics. There will be farmers who can still walk over the border from nowhere with a bag of amoxicillin and stick it in their feed. It is very hard to stop that. Given the previous number of people coming across the border, we know that will still happen*” (KI3, animal health). 

### 3.4. Feedback

Although the original HKSAP outlined a simple monitoring and evaluation (M&E) plan, the extent of its implementation remains vague. According to one informant from the HLSC, to monitor the progress of the HKSAP, “*the HLSC held a meeting once per year […] The EC and the AMR working group also met at least two times a year. They were really talking about the work completed, results, and what is to be done*.” In addition, referring to the Mid-Term Review Report, the informant commented that “*I think the report is very detailed and has all the actions laid down in the plan (e.g., which stage we are on and what is missing). The EC actually gave lots of advice on that*” (KI8, human health). 

For the animal health sector, representatives of AFCD in committees and working groups related to AMR listened to comments and feedback on the status of AMR in the sector. Afterward, one person would collate the information and bring it back to AFCD for further discussions. A human health informant confirmed the existence of this feedback loop. “*[In an HLSC meeting], I mentioned that as our system changes, micro-organisms are not enough. We should also look at AMR genes and the transfer between genes. The AFCD did start to look at the AMR genes in animal products*” (KI8, human health). However, it is unclear if there were similar practices in the human health sector. 

### 3.5. Interactions and Tensions

Overall, intersectoral and cross-sectoral collaboration has gradually come together during the implementation process of the HKSAP. Government departments, academic institutions, and industry associations engaged with one another in a meaningful way. For the animal health sector, three informants mentioned that farmers in Hong Kong have close relationships with the veterinarians, the AFCD, and the CityU. In one informant’s opinion, “*it is working out very well. It is a very good example of the government, the university, and the farmers all working together*” (KI10, animal health).

Furthermore, HLSC-led meetings became the major avenue for cross-sectoral collaboration. At these meetings, representatives from each sector would convene to express their views and experience on the situation and problems during the implementation process. “*Among the government, I think there is good coordination between the government departments, officials, and medical professionals. They are really talking about the work, the results, and what needs to be done*” (KI8, human health). One informant who worked for both HA and an academic institution shared their personal experience in multisectoral cooperation: “*When I was in Hospital Authority, we had to deal with the universities, and I am now in university. Another critical player is the Centre for Health Protection. We are also working closely together*” (KI6, human health).

However, some informants still viewed the multisectoral and intersectoral coordination as weak. This is because these informants were frustrated by how different sectors have been seen passing blame on certain issues. As an informant summarized, “*[HKSAP] is working for all of them, but they are not working with each other*” (KI1, DH official). The long-existing tension between the human health sector and the animal health sector offers a possible explanation for this problem. As reiterated by an animal health informant, “*it has always been for at least 50 years, a human versus animal tension. It is not just AMR. It is everything. The fingers always point at the agriculture sector as being the main culprit*” (KI3, animal health). Interestingly, human health informants also found the implementation of the HKSAP somewhat off-putting, as it disproportionately targeted the human health sector. As one informant said, “*I do not think the particular focus on human health would be too useful. […] We should not be pressured too much*” (KI6, human health). 

Our interviews further revealed that some of the higher-level policymakers were unfamiliar with the situation at the ground level and did not give enough autonomy to policy implementers at lower levels. Consequently, the HKSAP was perceived as inflexible by some informants. An informant argued that “*we need flexibility. You should have trust in the professionals working at the local level and let them tell you what is going to work and what is not*” (KI3, animal health). 

Unlike the strong ties between the government authorities, CityU, and farmers in the animal health sector, the alignment among cross-sectoral stakeholders was far from satisfying in the human health sector. The collaboration between the CHP and industry associations was reported to be very weak. For instance, “*[The CHP] never ever came to us and talked with us or had meetings. To be honest, there is no connection*” (KI5, human health). The coordination between the public and private healthcare sectors has also been problematic. The DH and other related government authorities did not contact private healthcare institutions to provide necessary guidelines or to involve them in other activities as they did with public hospitals. One informant from HA recognized that their communication with the private sector was lacking: “*We do not have any separate committees with a private hospital. A private representative from one private hospital is also a member of the EC. In that occasion, we can share. Otherwise, we do not have any personal or non-official communication on AMR*” (KI11, human health).

### 3.6. Environmental Factors

The implementation of HKSAP was bounded and supported by a complex set of environmental determinants. We found that international organizations’ advocacy for AMR policies, changes in the framing of AMR, and the COVID-19 pandemic were the most important contextual factors affecting the implementation of the HKSAP. 

#### 3.6.1. Political Factors

International organizations have been important in driving the adoption of the HKSAP. These organizations’ call to action on AMR and their newly launched projects have also been a major impetus for the implementation of AMR policies. An informant concisely summed up this point by saying: “*I think it is like an international obligation. WHO and OIE [World Organization for Animal*
*Health] are pushing towards this strategy. Hong Kong is also a part of the system. So, we need to play a role*” (KI12, human health). An informant recalled that “*AMR [policies] […] started with some high-level discussions. Then, at the bureau level. Then, the World Health Assembly mandated the AMR as the global emergency and Hong Kong launched a region-wide program*” (KI6, human health). The involvement of the WHO has gone further than just facilitating awareness. The organization actually “*inquired into the area and asked why we consume so many antibiotics in Hong Kong*” (KI5, human health), which may have exerted additional pressure on the Hong Kong government.

Informants also pointed to two internal factors that affected the implementation of the HKSAP. First, AMR has been gradually framed as more of a political issue in Hong Kong than as a health problem (KI2, DH official). Second, the governance structure in Hong Kong favors the implementation of the One Health concept and the HKSAP. According to an informant, “*Hong Kong has been lauded as a governance system that solves the problem because the Secretary of Food and Health oversees both the human health and agriculture sectors. I cannot think of any jurisdictions with the same arrangement*” (KI3, animal health).

#### 3.6.2. Economic Factors

The informants unanimously agreed that the reliance on imported products, especially food from mainland China, was a challenge for implementing the HKSAP (KI11, human health). Indeed, although local farmers are always incentivized to use antibiotics to reduce economic loss due to infectious diseases, it was never a significant factor, as “*most of the meat products or fishery products are imported from outside.*” (KI8, human health). However, for food traders, whether the imported products contain antibiotics was not a primary consideration, and profits always came first. 

#### 3.6.3. The COVID-19 Pandemic

The COVID-19 pandemic negatively affected the implementation of the HKSAP, with most implementing organizations prioritizing COVID-19 over AMR: “*it will take some time for AMR to regain its prominence in terms of policy*” (KI4, human health). One HA informant mentioned that “*our time to deal with the AMR has decreased due to COVID-19*” (KI12, human health). The DH had reportedly reallocated manpower and other resources initially designated for the HKSAP to handle the pandemic, which could have postponed the planned interventions and projects. Further, in some cases, people used COVID-19 as an excuse to justify delayed responses to requests from other departments. As mentioned by an informant, “*they always say, ‘we are so busy with COVID-19.’ Then, we have to wait*” (KI1, DH official).

However, the pandemic also had some positive effects on the HKSAP’s implementation. Many informants believed that during the pandemic, Hong Kong residents maintained good personal hygiene habits, and hospitals had adopted stricter infection control standards and measures, all of which were outlined in the HKSAP as strategic interventions. Moreover, COVID-19 “*offers a very good opportunity to do more healthcare-related promotion*” (KI7, human health). According to several animal health informants, the pandemic had increased awareness of biosecurity. As such, “*the COVID-19 is something that adds water into the fire*” (KI9, animal health). Moreover, a DH informant reiterated that “*COVID-19 would probably be less enduring than AMR. COVID-19 will go away eventually. But AMR will be here to stay for a long time*” (KI1, DH official).

## 4. Discussion

Based on interviews with policymakers and implementers from government bodies, industry associations, and academic institutions, our study found that important progress has been made in implementing the HKSAP, as evidenced by the intensified awareness-raising and educational activities, strengthened AMR surveillance, expanded infection control measures, and the promotion of the One Health concept. However, implementation remains suboptimal due to a complex set of reasons, such as cross-sectoral tensions, scattered data collection activities, weak enforcement of AMU regulations, and an entrenched perception of AMR. Some of these challenges were found to be linked to the weaknesses of the HKSAP (e.g., unclear accountability framework, lack of inputs from certain sectors during the policy design phase), making it less feasible to carry out all planned activities. 

Similar to previous studies, we found that AMR policy implementation is an inherently complicated and intricate process [41] requiring the active participation of cross-sectoral stakeholders at different levels. The overall implementation of the HKSAP was similar to the "vertical approach" adopted in England and France: setting goals at the central level (HLSC) with the advice of professional committees (EC), then implementing the policies through various agencies (e.g., CHP, AFCD, CFS) [42]. The HLSC and EC, in fact, had connections with different institutions and individuals. However, they were unable to bridge the gap between high-level policymakers and the needs of middle-level and frontline implementers. In addition, Hong Kong’s public health system struggles with “separating management from operation” [43]: The Food and Health Bureau (FHB) is responsible for the formulation of health policies, financial allocation, and supervision, while the HA and DH are in charge of the provision of services. Although this governance structure is conducive to fostering accountability, it creates barriers preventing the integration of the interests and perspectives of government agencies involved in the implementation of AMR policies, leaving out non-government actors, and this oftentimes leads to suboptimal policy implementation. The "diffuse governance model", which co-produces governance by a wide range of actors instead of having a dominated governmental agency alone [44], may be a potential option for Hong Kong to consider. In fact, France has already shifted to a "top-down" approach, using a more integrated multi-level governance approach on the AMR issue [42].

Optimizing AMU is a centerpiece of the HKSAP, but the planned activities have not been fully implemented or failed to produce the expected outcomes due to multiple constraints. As for the animal health sector, Hong Kong is highly reliant on imported food and animal products, where administrators have found it hard to control AMU in imported food and agricultural products, as well as stop local farmers from purchasing drugs across borders. Given that trade and AMR intersect across different sectors, regulations on imported commodities related to AMU deserve greater attention in Hong Kong. Some scholars have argued that strengthening surveillance and using standardized methods for data collection could be useful tools and, in turn, promote experience-sharing efforts as well as improve food trade security at the regional or even global level [45]. Others have suggested reinforcing legislation and enforcement policies to promote responsible AMU in veterinary services and food production, along with promulgating regulations on food safety and substandard medicines as part of trade agreements [46,47].

Obtaining public trust and policy "buy-in" by the target groups is a long-term process. Although tightening AMR-related regulations is important, measures with active behavioral changes are equally important. Our study provides concrete evidence that Hong Kong people’s preference for antimicrobials has deep-seated roots, while local farmers have economic incentives to purchase cheap and illegal antimicrobials. Furthermore, inappropriate medical prescriptions are still common, especially among community pharmacists. Thus, changing knowledge and attitudes towards AMR is essential for reversing the trends in AMR [48]. The HKSAP specifically emphasized this aspect, and many informants believed that educational activities for targeted groups were effective to some extent. However, we found that there was a lack of evidence detailing how these activities would be continuously organized, which could compromise their long-term effectiveness, a problem that has been reported in many other contexts [49]. 

Ideally, the One Health approach should bridge gaps between different sectors and foster communication and collaboration among sectors that “for too long operated in isolation” [50]. However, consistent with the existing literature, our study shows that translating the One Health concept into action was not straightforward in Hong Kong. The interview findings suggest that poor coordination was not only identified among different government agencies but also between government and non-government stakeholders. Within the human health sector, public hospitals received the most attention, and resources were disproportionately allocated to them compared to their private counterparts. Additionally, some private hospitals and clinics were not included in the current AMR data and surveillance systems. As 70% of outpatient services in Hong Kong are delivered in private healthcare settings [51], the inadequate engagement of the sector will limit the effectiveness of AMR policies. This finding conforms to the results of Dar et al.’s study, which also found evidence that AMR policy implementation was suboptimal in the private sector [52]. 

Other than the tensions within a specific sector, many informants expressed genuine concerns over the fragmented and imbalanced multisectoral collaborative model. The tension between the animal health and human health sectors was particularly pronounced. Similar to findings from Tanzania [16], Thailand [53], and Malaysia [54], where the implementation of NAPs was relatively inadequate in the animal husbandry, fisheries, and environmental sectors, AMR surveillance and control in the human health sector was stressed the most. The related implementation tools were also more developed in this sector compared to its counterparts. Our study unveils several possible explanations for this challenge: (1) unequal participation in policymaking; (2) weak enforcement of the One Health concept; (3) the tendency to prioritize easily accomplished goals; (4) the long-lasting “blame game” regarding which sector is the main contributor of the AMR problem in Hong Kong.

As different sectors in Hong Kong presently do not have coordinated governance structures that work in tandem, attempts to combat AMR have been fragmented, resulting in resource duplication and inefficiency [55]. More importantly, the weakly established M&E system of the HKSAP meant that the identified intersectoral and cross-sectoral tensions were not communicated to decision-makers for future modification in a timely manner. Our study thus revealed the missed opportunity to utilize M&E activities to facilitate cross-sectoral coordination. Moreover, the government should consider further investing in and building a surveillance system that is sustainable and systematic, allowing for the longitudinal collection of data in the animal health sector, and one can integrate these data with that of the human health sector. The Danish Integrated Antimicrobial Resistance Monitoring and Research Program is a prime example [56]; through this program, a One Health AMR surveillance network was established to further facilitate actions curbing the development of AMR.

As the Hong Kong government is concluding and evaluating its first HKSAP, the sustainability of such an action plan is a pertinent issue. For a well-resourced context such as Hong Kong, funding for AMR interventions was not as limiting as non-financial resources. There is a strong call for expanding the veterinary services workforce and channeling greater investment towards farm modernization in the animal health sector. Regarding the human health sector, stakeholders had requested more technical support to conduct AMR surveillance and increased manpower for infection control and clinical audits. During the COVID-19 pandemic, the enactment of the HKSAP was substantially disrupted. Nevertheless, opportunities for change were also identified, especially for efforts to raise the awareness of the target groups of the HKSAP towards AMR.

To the best of our knowledge, this is the first qualitative research study in Hong Kong examining the progress and challenges of the implementation of the HKSAP. Although our interviews were thorough and we assessed the information comprehensively, our study is not without limitations. First, our response rate was low; as a result, the sample size was relatively small, with none of the informants coming from the environmental health sector, despite substantial efforts from the team to recruit interviewees from this sector. Thus, the reported findings may not holistically reflect how the HKSAP has been implemented, and bias may exist, although we suspect that there has been less engagement from the environmental sector in AMR policy in Hong Kong. Second, despite our many attempts during the data collection phase, some government agencies with a significant stake in the HKSAP (e.g., the Drug Office of Hong Kong) could not be contacted and were not involved. Third, the findings from interviews are not easily generalizable to other locations, as this study was closely related to the Hong Kong context. Therefore, future research endeavors should collect additional data on antimicrobial use and surveillance in each sector, as well as engage more policy stakeholders to complement qualitative data, better understand how to address long-standing challenges, as well as determine how to further implement the HKSAP and other AMR policies in Hong Kong.

## 5. Conclusions

Using the Smith Policy Implementation Process Model, this study examined the implementation of the HKSAP, the progress made, and the associated challenges since it was first announced in 2017. While early evidence demonstrated promising changes in the local AMR landscape, notable challenges remain to be resolved. Interviews with policy informants showed that insufficient data surveillance, unclear accountability framework, lack of input from certain stakeholders, and misalignment when it comes to multisectoral collaboration are major implementation challenges that have prevented the HKSAP from realizing its stated objectives. Looking ahead, facilitating meaningful and continuous engagement with professionals and the public, strengthening data collection and surveillance under the One Health concept, and enhancing inter- and cross-sectoral collaboration should be prioritized for more effective implementation of the HKSAP.

## Figures and Tables

**Figure 1 antibiotics-11-00636-f001:**
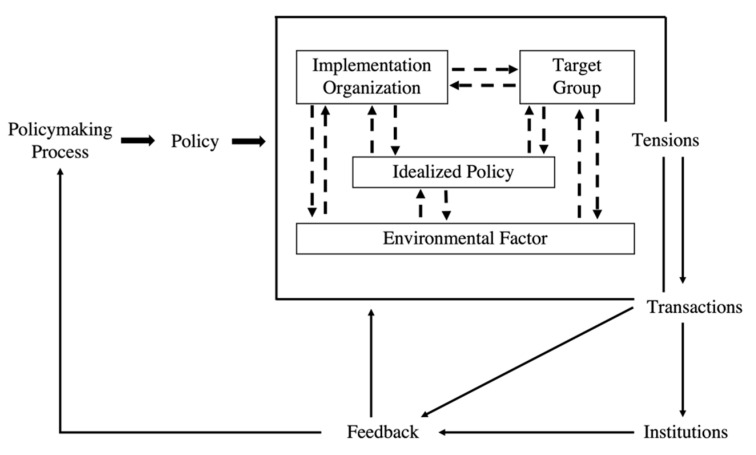
The Smith Policy Implementation Process Model (recreated by authors).

**Table 1 antibiotics-11-00636-t001:** Examples of interview questions.

Theme	Question
Views on the implementation of the HKSAP	What are your general views on implementation of the HKSAP in Hong Kong? How do you think the issue with AMR has changed over the past few years? What are the main areas that require attention? Which sectors are actively involved in implementing the HKSAP over the past few years?
AMR surveillance	Please describe the existing surveillance system to combat AMR in your institution. (Prompts: Which areas are done well? Which areas require further action?)
Education and public awareness	What are the existing strategies for improving awareness and understanding of AMR? (Prompts: Has the government tailored these efforts according to the target audience? Is the effectiveness of these AMR awareness campaigns/activities being measured?)

**Table 2 antibiotics-11-00636-t002:** Sociodemographic description of all participants.

	Count
Sex	
Female	6 (35%)
Male	11 (65%)
Sector	
General	2 (12%)
Human health	11 (65%)
Animal health	4 (23%)

## Data Availability

Due to the risk of identificatication of individuals or organizations, participants of this study did not agree for their data to be shared publicly, so supporting data is not available.

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
