# Peer review of "Understanding the Implementation of Antimicrobial Policies: Lessons from the Hong Kong Strategy and Action Plan"

_antibiotics, 2022, doi:10.3390/antibiotics11050636_

Round 1

Reviewer 1 Report

The manuscript submitted by Song et al on "understanding the Implementation of Antimicrobial Policies Lessons from the Hong Kong Strategy and Action Plan" highlights the significance of AMR and antimicrobial stewardship policy implementation in Hong Kong. AMR is a major global crisis, and it is good to know  Hong Kong has HKSAP Hong Kong and a strategic action plan for AMR. The work involves a collection of views in the form of consent interviews from 17 govt officials involved in HKSAP and identified various factors affecting the implementation of the above. In light of the current crisis, I would very much welcome this approach.

My comments are:

The manuscript is well written and relevant to the 'Antibiotics’ journal.

The agricultural sector is a significant contributor to the AMR crisis, and I would see more interviewees from this sector. I like the authors' point that the farmer's illegal antibiotics purchase from borders and inappropriate usage to increase their production.

In terms of participants, the number is meagre. I would add more rationale for this study, which would be beneficial for a wider audience or other countries' policymakers rather than Hong Kong people. It would be good to see more on one health approach status in Hong Kong, which would be a good addition to this manuscript.

Author Response

Dear reviewer,

We would like to thank you for taking the time to evaluate our manuscript. We have revised the article based on your feedback, and our responses to each comment are presented below.

Comment 1: The agricultural sector is a significant contributor to the AMR crisis, and I would see more interviewees from this sector.

Response 1: We agree with the reviewer that the agricultural sector plays an important role in the AMR problem – as well as the solutions. However, we made substantial efforts to recruit potential informants from the sector from April to June 2021. We searched the contact directory of the agriculture department and sent out tailored invitations to key officers which could have involved in handling AMR or implementing the HKSAP. Moreover, we asked interviewees for their recommendations on study participants/direct referral at the end of the interview. On average, each contact in our list was reached out by the research team for two times. However, we were unable to secure interviews from this sector. We believe that this reflects the fact that this sector has not been as engaged in the AMR process in Hong Kong. We have added more discussion in our limitations section about the lack of inclusion of the environmental sector in our study.

Comment 2: In terms of participants, the number is meagre.

Response 2: Similar to our response to the first comment, we tried different approaches to involve as many informants relevant to the research topic as possible. Nonetheless, as the government and the non-government sectors were occupied by COVID-related activities, the majority of potential interviewees did not reply to our emails or declined to participate in the study. In addition, Hong Kong has a relatively small group of AMR-focused professionals. Due to these factors, we believe that the current number was acceptable. We recognized this as a major limitation of our study in the “Discussion” part. Meanwhile, we also heavily used direct quotes from informants to indicate that his/her opinion or the example mentioned in the quote may not be generalized to other sectors or organizations.

Comment 3: I would add more rationale for this study, which would be beneficial for a wider audience or other countries' policymakers rather than Hong Kong people.

Response 3: We thank the reviewer for this suggestion. We added a sentence to the last paragraph of the “Introduction” part to describe the rationale/significance of our study. 

Comment 4: It would be good to see more on one health approach status in Hong Kong, which would be a good addition to this manuscript.

Response 4: We appreciate that the reviewer brings up this point. In the “Introduction” section of the manuscript, we briefly summarized the issue landscape and policy developments related to AMR. We did not highlight the One Health approach because there were limited literature to inform our analysis. Another reason was that we aimed to write a policy implementation paper instead of merely focus on AMR.

However, we agree with the reviewer that the One Health concept is a core element of every AMR action plan. So, we addressed this aspect in “Findings” and “Discussion”, where the status of intersectoral collaboration and coordination between stakeholders within a sector were thoroughly discussed.

We look forward to hearing from you in due time regarding our submission and to responding to any further questions and comments you may have.

Sincerely,

Karen Ann Grépin

Reviewer 2 Report

The manuscript covers an interesting and up-to-date topic of implementation of antimicrobial resistance policy in Hong Kong. The article is of high merit, however, some minor issues must be corrected to make the whole article more clear for readers.

General:

-Authors should pay attention to text formatting, as there are some double spaces (e.g. in the Introduction).

Abstract

-Each abbreviation must be explained when it is used for the first time – please explain what “AMR” means.

Introduction

-Please present some background about antimicrobial resistance as a first paragraph in the Introduction – what is the etiology of this problem? I would also suggest to add some data concerning the range of the problem of antimicrobial resistance – are there any differences in the prevalence in different countries/regions?

-Once again, each abbreviation must be explained – what does “NAPs” mean?

-In paragraph 2 Authors are talking about different countries which focused on factors underlying AMR policy implementations – please indicate what these countries are (references 6-8).

Materials and methods

-It’s written that Authors used semi-structured interviews. If so, what were the exemplary questions which were asked? (as I understand interviewers did not strictly follow a formalized list of questions?)

-I think that Authors should add the mean age of the interviewees in Table 1.

Discussion and Conclusion

-I find the Discussion interesting and of high merit. Conclusion summarizes the whole manuscript, as well as highlights further  actions for improvement of the HKSAP.

Author Response

Dear reviewer,

We would like to thank your for taking the time to evaluate our manuscript. We have revised the article based on your feedback, and our responses to each comment are presented below.

General comments

Comment 1: Authors should pay attention to text formatting, as there are some double spaces (e.g. in the Introduction).

Response 1: Thank you for pointing this out. We went through the manuscript carefully in this round of edit to ensure all text formatting issues are resolved. 

Abstract

Comment 1: Each abbreviation must be explained when it is used for the first time – please explain what “AMR” means.

Response 1: Agree. We added “antimicrobial resistance” in front of “AMR” when the abbreviation was used for the first time.

Introduction

Comment 1: Please present some background about antimicrobial resistance as a first paragraph in the Introduction – what is the etiology of this problem? I would also suggest to add some data concerning the range of the problem of antimicrobial resistance – are there any differences in the prevalence in different countries/regions?

Response 1:  You have raised an important point here. We have incorporated your suggestion by adding a paragraph at the beginning of the article. We articulated the fatality of AMR and cited the latest data on AMR burden. We also mentioned the disparities across regions.

Comment 2: Once again, each abbreviation must be explained – what does “NAPs” mean?

Response 2: Agree. We fixed this issue by adding the full name of NAP to where it firstly appeared in the paragraph. We also checked other abbreviations in this article and provided explanations (if applicable).

Comment 3: In paragraph 2 Authors are talking about different countries which focused on factors underlying AMR policy implementations – please indicate what these countries are (references 6-8).

Response 3: Revised accordingly.

Materials and methods

Comment 1: It’s written that Authors used semi-structured interviews. If so, what were the exemplary questions which were asked? (as I understand interviewers did not strictly follow a formalized list of questions?)

Response 1: Thank you for the suggestion. We added a statement in the Method section indicating that the interview questions for each informant were tailored based on their background, expertise, sector where they have been working, and their engagement with the HKSAP. We created a table below this paragraph where we listed some exemplar interview questions.

Comment 2: I think that Authors should add the mean age of the interviewees in Table 1.

Response 2: Thanks for your comment. We are aware that some qualitative studies do report the median age of the interviewees, however, we did not collect this information at the time of our interviews.  In our study, our interviewees are all professionals in their respective fields, and thus there was not that much variation in the sample as most were in the mid-40s to mid-50s.  Plus, given that the number of interviewees participated in this study is quite small, we think that providing the data added limited value in terms of showing the representativeness of interviewees. As these people were speaking in their roles as professions, we believe that age would have limited effects on our research findings.

We look forward to hearing from you in due time regarding our submission and to responding to any further questions and comments you may have.

Sincerely,

Karen Ann Grépin